# The Effect of the Video Assistant Referee System Implementation on Match Physical Demands in the Spanish LaLiga

**DOI:** 10.3390/ijerph19095125

**Published:** 2022-04-22

**Authors:** José C. Ponce-Bordón, David Lobo-Triviño, Ana Rubio-Morales, Roberto López del Campo, Ricardo Resta, Miguel A. López-Gajardo

**Affiliations:** 1Faculty of Sport Sciences, University of Extremadura, Boulevard of the University s/n, 10003 Cáceres, Spain; joponceb@unex.es (J.C.P.-B.); davidlobo123@gmail.com (D.L.-T.); anarubmor94@gmail.com (A.R.-M.); 2LaLiga Sport Research Section, 28043 Madrid, Spain; rlopez@laliga.es (R.L.d.C.); rresta@laliga.es (R.R.)

**Keywords:** decision making, match analysis, physical performance, professional soccer, video-replay technology

## Abstract

The present study aimed to analyze the influence of the Video Assistant Referee (VAR) on match physical demands in the top Spanish professional football league. Match physical demand data from all the matches for two seasons (2017/2018 and 2018/2019) in the First Spanish Division (*n* = 1454) were recorded using an optical tracking system (ChyronHego^®^). Total distance, relative total distance covered per minute, distance covered between 14–21 km·h^−1^, distance covered between 21–24 km·h^−1^, and distance covered at more than 24 km·h^−1^ were analyzed; also, the number of sprints between 21–24 km·h^−1^ and more than 24 km·h^−1^ were taken into consideration. The times the VAR intervened in matches were also taken into account. Results showed that total distance and relative total distance significantly decreased in seasons with VAR compared to seasons without VAR. Finally, distance covered between 21–24 km·h^−1^, distance covered at more than 24 km·h^−1^, and the number of high-intensity efforts between 21–24 km·h^−1^ and more than 24 km·h^−1^ increased in seasons with VAR compared to seasons without VAR, but the differences were nonsignificant. Thus, these findings help practitioners to better understand the effects of the VAR system on professional football physical performance and to identify strategies to reproduce competition demands.

## 1. Introduction

Research has shown that referees must make more than 130 observable decisions during football matches [1]. The increasing development and progress of technology in football supports and helps referees make better, unbiased decisions [2]. One of the technological advances is the Video Assistant Referee (VAR) system. The presence of the VAR system is increasing in elite football worldwide in order to minimize referees’ erroneous decisions and thus reduce subjective error when an action is unclear or needs to be rechecked [3]. After being officially introduced in the 2018 World Cup, major leagues started using the VAR as of the 2018/2019 season [4]. Therefore, the introduction of the VAR system may affect professional football, so research has recently analyzed its influence on team performance and play in elite football.

Some studies have analyzed the effect of the VAR on different football statistics. Specifically, Lago-Peñas et al. [5] examined the influence of the VAR system in 1024 matches of two professional European leagues (Italian Serie A and German Bundesliga) across the 2016/2017 season (without the VAR system) and the 2017/2018 season (with the VAR system). They reported a decrease in the number of offsides, fouls, and yellow cards after the implementation of the VAR. There was also an increase in the number of minutes added to the total playing time in the first half and the full game (i.e., extra time in playing total time). In the same vein, Han et al. [6] analyzed the Chinese Super League (CSL) during the 2017/2018 and 2018/2019 seasons to compare the influence of the VAR system implementation. These authors found a significant increase in the total playing time of both the first and second half, as well as for the whole match; and the number of offsides and fouls dropped significantly. Gurler et al. [7] showed that, in the Turkey Super League, average match time (seconds) and the ball in playing time (seconds) per game increased after the use of the VAR. Similarly, Lago-Peñas et al. [8] reported a significant increase in the number of minutes added to the playing time in the full match after the implementation of the VAR. However, the possible disruption to the flow and pace of the match due to the stopping and starting, which can be especially disruptive, has been criticized [9].

Although some research has been carried out on the influence of the VAR system implementation on football, these studies have not analyzed the influence of the VAR system on match physical demands. Only two studies analyzed the impact of VAR technology on physical performance. Firstly, a total of 375 matches of Spanish LaLiga during the 2018/2019 season were examined according to the number of VAR interventions per match, finding a slight decrease in the total distance (TD) covered [10]. However, these authors concluded that the intensity and rhythm of play were somewhat higher in matches where the VAR intervened. The disruptions of the match caused by VAR influence the effective playing time [9]. This has not yet been analyzed but players can use them to recover and perform more high-intensity efforts [11]. Secondly, Costa-Oliveira et al. [12] analyzed the average distance covered by the teams, the number of individual and total sprints per team of 116 football matches (57 matches of the 2014 and 59 matches of the 2018 FIFA World Cups). The results showed a significant increase in average distance covered and a significant decrease in the number of individual and total sprints during the tournament in which the VAR was employed. But the authors suggested prudence because these results could be related to other factors, such as match status or opponent quality. Therefore, the significant differences found between the two tournaments (i.e., 2014 and 2018 FIFA World Cups) should not be exclusively or directly attributed to the VAR system implementation, and any attempts to generalize findings or extrapolate inferences requires caution.

The studies presented thus far reveal opposite results and scarce evidence to establish clear conclusions about the influence of the VAR system on match physical demands. The external load has evolved over the last years, increasing the number of high-intensity actions [13]. Moreover, players can use the VAR disruptions to recover and perform more high intensity running actions [11]. However, whether the VAR system implementation strongly affects match physical demands, specifically high-intensity efforts in the First Spanish Division, is still uncertain. Therefore, this study aimed to analyze the influence of the VAR system implementation on match physical demands in the top Spanish football league across the 2017/2018 season (without the VAR system) and the 2018/2019 seasons (with the VAR system). Based upon prior studies [10], it was hypothesized that (i) total distance covered by teams would decrease in seasons with the VAR system implementation and (ii) distances covered at high-intensity and sprinting actions would increase in seasons with the VAR system implementation.

## 2. Materials and Methods

### 2.1. Participants

The sample included observations of all the matches played over two seasons (from 2017/2018 to 2018/2019) in the First Spanish Division (LaLiga Santander; *n* = 1454). Two observations were made per match, and one per team. Goalkeeper activity was included. Specifically, a total of 1454 out of 1520 potential records were included in the study. A total of 66 observations were excluded due to technical problems in the data collecting system or adverse weather conditions during the match. As the VAR was introduced in top European leagues at the start of the 2017/2018 season, VAR interventions were only considered in all the matches of the 2018/2019 season in the First Spanish Division. Data were provided to the authors by LaLiga^TM^, which had informed all participants through its protocols. All data were anonymized according to the Declaration of Helsinki to ensure players’ and teams’ confidentiality, and the protocol was fully approved by the Ethics Committee of the University of Extremadura; Vice-Rectorate of Research, Transfer, and Innovation-Delegation of the Bioethics and Biosafety Commission (Protocol number: 239/2019).

### 2.2. Design and Procedure

Match physical demand data were collected by an optical tracking system, ChyronHego^®^ (TRACAB, New York, NY, USA). This multi-camera tracking system assesses the distance covered in meters by teams and the number of high-intensity sprints (LaLiga™, Madrid, Spain). It consists of eight super 4K-High Dynamic Range cameras based on a positioning system (Tracab—ChyronHego video-tracking system). This system records and analyzes X, Y, and Z positions for each player from several angles, thus providing real-time three-dimensional tracking (tracking data are recorded at 25 Hz). This instrument is also based on the correction of the semi-automatic video-tracking system (the manual part of the process). The validity and reliability of the Tracab^®^ video-tracking system have also been recently tested for physical performance, reporting average measurement errors of 2% for physical performance variables [14]. In addition, recent studies have tested the agreement between the Mediacoach^®^ system and GPS devices [15]. Specifically, the magnitude of the intraclass correlation coefficients (ICC) was higher than 90 [15].

### 2.3. Study Variables

Following previous studies [16,17], match physical demands by teams were recorded by Mediacoach^®^ in different speed ranges: total distance covered by football teams in meters (i.e., TD), relative total distance covered by football teams per minute (TD/min.), distance covered between 14–21 km·h^−1^ (Medium-intensity running distance = MIRD), distance covered between 21–24 km·h^−1^ (High-intensity running distance = HIRD), and distance covered at more than 24 km·h^−1^ (Very high-intensity running distance = VHIRD). Additionally, the number of high-intensity efforts performed was also divided into two speed ranges: number of very high-intensity running efforts between 21–24 km·h^−1^ (Sp21); and number of sprints at speeds above 24 km·h^−1^ (Sp24). All of these variables constitute the total sum of the physical performance of the players who participated in each match (i.e., all the players who completed entire matches, all the players who were replaced, and the substitute players).

### 2.4. Statistical Analysis

Data were analyzed using the statistical program SPSS 25.0 (Armonk, NY, USA: [18]). Descriptive statistics for each variable in the First Spanish Division during two seasons (2017/2018 and 2018/2019) are presented as means and standard deviations. The Levene test was applied to check the equality of variances, and the Kolmogorov Smirnov test was used to establish data normality [19]. Firstly, a two-way analysis of variance (*ANOVA*) was used to explore the main differences in the professional football league for external load variables (i.e., variables related to distances covered and the number of sprints) across both seasons with and without the VAR. A post hoc comparison between the seasons, using Bonferroni posthoc analyses, was carried out. Thus, *ANOVA* [20] revealed the differences between seasons with and without VAR in match physical demands, where the football season was the factor. Finally, the standardized differences, or effect size (ES, 90% confidence limit), for the selected variables were calculated. We used threshold values for Cohen’s ES statistics of trivial (<0.2), small (0.2–0.6), moderate (0.6–1.2), large (1.2–2.0), and very large (>2.0; [21]). Statistical significance was set at *p* < 0.05.

## 3. Results

Table 1 shows an overview of the mean match physical demands comparison of the First Spanish Division between the two league seasons (i.e., with and without the VAR). The data show that, in this competition, TD and TD/min. significantly decreased (*p* < 0.05), in seasons with the VAR compared to seasons without the VAR. Non-differences were found between the two seasons in MIRD covered by teams. Furthermore, non differences were found in the distance covered at high-intensity, such as HIRD and VHIRD, and the number of high-intensity sprints, like Sp21 and Sp24, in seasons with the VAR system implemented compared to seasons without the VAR.

## 4. Discussion

The VAR system was implemented in football games in 2018 to support referees’ decision-making processes and to avoid possible match-changing incidents. Research has suggested that referees often make decisions that could directly influence the final score [22]. However, scarce evidence has been reported about the influence of the VAR system on match physical demands [10,12]. Therefore, this research aimed to assess the influence of the VAR system implementation on match physical demands in LaLiga Santander. The main findings of the study were as follows: (i) TD and TD/min. significantly decreased in seasons with the VAR compared to seasons without the VAR in the First Spanish Division; and (ii) a positive trend was reported in distances covered at high intensity and the number of high-intensity efforts in the 2018/2019 season (with the VAR system implementation).

Regarding TD covered by teams, we expected that it would decrease in seasons with the VAR system implementation (Hypothesis 1). The most striking result to emerge from the data is that TD and TD/min were significantly reduced in seasons with the VAR compared to seasons without the VAR. Thus, the VAR system implementation has involved a decrease in the TD covered by soccer players. Similarly, Errekagorri et al. [10] also observed a slight decrease in the TD covered by the First Spanish Division teams when the VAR system intervened. There are several possible explanations for this result. First, the effective playing time may have decreased when the VAR intervened and, consequently, the teams covered less TD [23]. Secondly, this result could be due to the natural evolution of match physical demands over the years, as research has reported that TD covered by teams of the First Spanish Division decreased significantly in recent years [13]. Third, the playing style used by the teams of LaLiga could influence these results, because in recent years there has been a gradual increase in teams with high ball possession levels, confirming that in ball control plays with few transitions, players covered less total running distances [24,25]. On the contrary, when World Cup matches were analyzed, the TD covered by teams was greater in championships when the VAR system intervened compared to championships without VAR interventions [12]. However, prudence is necessary when interpreting these data, as these results may be related to other factors, such as match status or the quality of the opposition, or even the playing style used by teams [26].

According to our second hypothesis, we expected an increase in the distances covered at high intensity and the number of high-intensity running efforts (Hypothesis 2). The results showed that distances covered at high-intensity, such as HIRD and VHIRD, and the number of high-intensity efforts, like Sp21 and Sp24, increased in seasons with the VAR compared to seasons without the VAR, although the differences were nonsignificant. Thus, the VAR system implementation produced a positive trend in the distances covered at high intensity by soccer players. This result may be explained by the fact that the VAR interventions could cause a decrease in effective playing time (i.e., there were more playing disruptions), which allows players to cover less TD and rest more often in order to perform more high-intensity efforts [10,11]. Similarly, recent research has shown that distances covered at high intensity and high-intensity efforts have increased in the last years in the First Spanish Division [13]. The increase in distances at high-intensity could also be an indicator of the evolution of football, where football players are now trained to perform more high-intensity actions [27]. It may also be a consequence of the playing styles used by teams of LaLiga, where it has previously been shown that there has been a gradual increase in teams with possession-play and that teams with possession-play performed more high-intensity sprints [25]. On the other hand, Costa-Oliveira et al. [12] reported a reduction in the number of individual and total sprints in the 2018 FIFA World Cup, where the VAR was implemented, compared to the previous edition (without the VAR system). There is some controversy in the findings of the current research, so our study aims to be more representative due to the included sample and to go further to understand the influence of the VAR on match physical demands.

### 4.1. Limitations and Future Perspectives

This study has some limitations that should be acknowledged with a view to future research. First, we should perform a longitudinal comparison over the years of football competition (i.e., more seasons), before and after the VAR system implementation, to obtain more representative and consistent data. Moreover, we did not analyze other physical variables such as accelerations and decelerations [28] and high metabolic load distance (HMLD; [29]), which are part of the external load of football matches and should be analyzed to gain more specific knowledge of the competition. Furthermore, a comparison between the First and Second Divisions regarding the influence of the VAR system on match physical demands could also be introduced. Moreover, an extrapolation of such a study to other European leagues could be included, so we could observe differences in the VAR system implementation between different football leagues. Finally, due to the VAR, there is more time in the minutes of the discount because a VAR decision can hold up the match for several minutes (i.e., the real “active” time—ball in action—during the matches decreases), so it would be interesting to analyze how the decrease of effective time caused by the VAR affects match physical demands.

### 4.2. Practical Applications

Some practical applications can be extracted from the results obtained. First, the VAR system implementation has gently changed the match physical demands across the seasons. In particular, the TD covered has significantly decreased and distances covered at high-intensity have shown a positive trend. These results provide very useful knowledge to strength and conditioning coaches to apply specific strategies for the distribution of external load in training sessions to reproduce competition demands and optimize players’ physical performance. For instance, strategies to prevent hamstring injuries such as eccentric strengthening and sprint training should be included during the season and, especially, in the pre-season and off-season [30]. These stimuli constitute a method for injury prevention and could reduce the injury rate of football players and allow them to perform a higher number of high-intensity efforts in soccer matches.

## 5. Conclusions

The present research represents an attempt to determine how match physical demands have evolved in professional football teams after the VAR system implementation. Overall, the findings of the present study showed that the VAR introduction in football has produced a significant decrease in TD and TD/min., and a positive trend was reported in distances covered at high-intensity and the number of high-intensity efforts in the 2018/2019 season. That is, football players covered less distance with the VAR. These findings suggest that the VAR system affects the game in elite football. The training strategies should provide stimuli for football players to perform greater high-intensity efforts in matches.

## Figures and Tables

**Table 1 ijerph-19-05125-t001:** Match physical demands comparison between seasons with and without the VAR in LaLiga Santander.

	Season 2017/2018			Season 2018/2019			*p*	∆*Cohen*
	*M*	*SD*	95% CI	*M*	*SD*	95% CI		
**TD (m)**	** 109,321 **	4189	109,013, 109,629	108,596	4497	108,289, 108,902	**	0.17
**TD/min. (m/min.)**	1505	74	1500, 1510	1477	73	1472, 1482	***	0.38
**MIRD 14–21 km·h^−1^ (m)**	22,709	2322	22,557, 22,861	22,471	2216	22,320, 22,622		0.10
**HIRD 21–24 km·h^−1^ (m)**	3013	384	2986, 3041	3056	396	3028, 3084		0.11
**VHIRD > 24 km·h^−1^ (m)**	2930	486	2894, 2965	2960	500	2925, 2995		0.06
**No. Sp 21–24 km·h^−1^**	264	30	262, 267	268	32	266, 270		0.13
**No. Sp > 24 km·h^−1^**	161	22	160, 163	162	23	160, 164		0.04

Note. SD = Standard Deviation, m = meters, No. = Number, TD = Total distance, MIRD = Medium-intensity running distance, HIRD = High-intensity running distance, VHIRD = Very high-intensity running distance, Sp 21–24 = Sprints between 21–24 km·h^−1^, Sp > 24 = Sprints at speeds above 24 km·h^−1^, ** *p* < 0.01. *** *p* < 0.001.

## Data Availability

Restrictions apply to the availability of these data. Data was obtained from LaLiga and are available with the permission of corresponding author.

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
