# Peer review of "The Effect of the Video Assistant Referee System Implementation on Match Physical Demands in the Spanish LaLiga"

_ijerph, 2022, doi:10.3390/ijerph19095125_

Round 1

Reviewer 1 Report

Thank you for the opportunity to review this manuscript. The purpose of the study was to investigate the influence of the VAR system implementation on match physical demands in the top Spanish football league, La Liga. A season (2017/18), wherein the VAR system was not used, as compared to a season (2018/19), where the VAR system was used.
I can see the interest of the topic for the readership of the scope of IJERPH.

The quality of your English needs to be revised throughout the manuscript. The draft needs to be revised by a native speaker.

General:

As you have submitted your manuscript for publication in a European Journal and did your research in a European Top League, I would recommend using the term football, as it is common in Europe. “Soccer” is isolated used in the North American hemisphere.

Abstract/Title:

Please, reduce the number of abbreviations in the Abstract. It’s pretty hard to follow with such plenty of abbreviations. Try to limit those to the relevant ones.

Exclude any statistical parameters from the Abstract.

Try to write your Abstract as plain and clear as possible. This should address all potential readers, without only addressing experts in this field. Therefore, the Abstract needs to be capable for everybody and not limited or distracted by too much unnecessary content. The details are explained in your manuscript.

L22-23: I do not see how this conclusive statement is supported by the results and findings of your study.

Manuscript:

Introduction:

General comments:

You need to better elaborate, on why it is important to investigate the changes of physical demands due to the implementation of the VAR system! You list reasonable arguments, but your Intro lacks a clear rationale for why there seems to be necessary research on the relationship of VAR and the physical demands in a football match. You need to better elaborate the purpose of your research. I suppose all relevant arguments and details are in your Introduction, but you should re-arrange them more logically and highlight, why you expect or not that the VAR has a significant influence on a player’s physical demands and that this has any superior influence on the overall game performance.

Specific Comments:

L28-42: You highlighted in your title the new Video Assistant Referee technique; in the first section of your Introduction, you, firstly, describe the physical changes in the game and how they are success-relevant with the necessity to control and analyze. That’s fine, although I did expect that you start with the VAR technology to introduce. But what I don’t get is, how you linked the tracking of physical demands of a game to the VAR technology. I do not see the link you have provided, because those systems are not fully associated, although using similar technology.

Methods

L104: Full affiliations of the system are missing here.

L103-106: Re-order in two distinguished sentences.

L124: Affiliations of SPSS are incomplete.

L132: I do think ‘investigate’ is not appropriate here. Maybe change to ‘revealed’ or similar.

L134-135: Why did you set p a priori at three different levels? And how did you apply those different levels to your data set or rather to the sub-research questions? Normally, you set p at one level for all data. Specific p-values (i.e., 0.01, 0.001, or else) are communicated at any single analysis anyways, right?

Results

L138-139: “What is interesting…” is an interpretative statement. Yet, the Results section should solely communicate the plain results of your study. Without any interpretation – therefore, the Discussion section is meant for.

L140: And: It shouldn’t matter if a result is p < .01 and p < .001. According to you’re a priori set p-level, results lower that threshold is significant; do not distinguish in categories like significant, more significant, or highly significant – because this makes content-related no sense. A difference is significant or not!

L141-142: If a decrease is not statistically significant you should be hesitant with defining it as a decrease. In scientific papers the stats tell you if there is a difference or not, also a meaningful decrease or increase; if not significant, it is not meaningful. This has to be elaborated more meticulously.

L142-146: Same here, as commented before. Also, try to create clearer statements in the Results, one result after another, and always give the inferential stats in parentheses after each statement. Do not merge too many results in one statement; just if they belong to one sub-research question.

Table 1: The Table appears very confusing. Please re-edit. Especially, give decimal places that make sense; I do not think with such high values two decimal places are necessary; even one could be discussable. Furthermore, re-arrange the rows and columns that are uniform and consistent in their appearance. Afterward, check your Table(s) that stand-alone criteria are fulfilled. Every Table (and Figure) needs to be self-explanatory, with all relevant content included and explained by the caption.

Figures: I would recommend the Figures at a position in the manuscript, where you cross-referenced those.

The y-axes appear to have quite weird and different values. I would recommend making this more consistent (where possible) that the Figures are easier to compare. But those classifications of the axes appear totally arbitrary and do not help to understand the Figures better, instead, at first glance all bars appear to have the same height. Furthermore, I miss titles and asterisks depicting significance where occurring.

Discussion

General: I suggest you could enhance the quality of your Discussion by a more in-depth discussion of the results and findings that became significant in your investigation. You are rather discussing the findings you have expected anyways. And, which supports the general development in professional football, with a higher occurrence of high-intensity loads. It would be good if you could point out more clearly what the new findings are due to your analysis.

L172-174: You shouldn’t state it like this when you couldn’t detect significance in the decrease. Weaken your statement; maybe talk about tendencies. And: Consider the calculation of effect size (eta2 and Cohen’s d), to get a better idea of the difference! That would highly improve the quality of your results.

L175-176: Why did you actually expect that? And, is this also valid, when analyzing the net playing time? Do you have any sentiment for that?

L215-223: It’s quite uncommon to point out the strength of your analysis at the end of the Discussion again. I just want to say, you pointed out the strength of your research and alongside your paper before, otherwise, you wouldn’t assume that its content is relevant for scientific publication, right? And, such a paragraph rather appears as a repetition of facts that already have been risen throughout the paper before. So, I would suggest removing this part.

L241: I assume this statement is a bit far out of your results and findings. You should modify that because this is a very strong statement. And, from my point of view, with your findings, you cannot generally state that.

L245: I am not totally convinced: If the players have a higher load in high-intensity situations in the game and if following your recommendation would also train in-between the game cycle also in a high amount high-intensity training loads, wouldn’t bear a high risk to totally exhaust the players, and with that would increase the risk of injury tremendously? Of course, the game changes necessitate professional steering of the training bouts also throughout the game cycle, but how about pre-season and off-season, where the main work towards building up a solid base of physical fitness is done? Maybe you could also include some practical implications towards that point of the training cycle.

L246-248: Once again, I am very hesitant to be convinced by that statement that if having more high-intensity game loads, training between two games should also include more high-intensity training loads to reduce injury risk! Where in this schedule is the player meant to fully recover? With that practical implication, I would assume the players cannot fully recover (even tough game rhythm is rather 3-4 days than 7 days) and the risk of injuries highly increases. That might also be an interesting question to solve if (muscular) injuries increased in recent years because of the change of gameplay’s intensity?

Conclusion

L254: I would frame the insignificant results rather as tendencies.

Thanks for your work and I would say that your draft is already in acceptable shape! I encourage you to put some more meticulous work into it and you will see the contribution and enhancement.

Author Response

Hi reviewer 1,

here are the pertinent revisions made to the manuscript.

Reviewer 2 Report

Dear authors.

You have presented an interesting work about performance factors on soccer about the intervention of VAR on the running performance during the games. However, the analysis of the seasons 2017/2018 and 2018-2019  are somewhat updated. How you express in the limitations, if you had included recent seasons the work would have been more interesting.

In my opinion introduction, method section and discussion are correct. You provided a enough content about the context of the manuscript and you have used interesting previous research under the same line that your research. 

I have only suggestion, which I consider that improve your work and will reinforce your results:

  • Maybe, is it possible to add information about “active” time in the season? This is an important variable. We know that with VAR there are more minutes in the minutes of the discount because a VAR decision could stop the match several minutes. However, in your analysis you don’t include the real “active” time (ball in action) during the matches. I consider that it is interesting if the active time is less with the VAR intervention. Maybe this is one of the reasons which explain that a total distance time and total distance/minutes has been reduced with the VAR apparition. You explain this in the discussion section. Maybe it would be interesting (with your big sample) analyze how affect the VAR to the effective time.

Best regards.

Author Response

Reviewer comments

Reviewer 2

Dear authors.

You have presented an interesting work about performance factors on soccer about the intervention of VAR on the running performance during the games. However, the analysis of the seasons 2017/2018 and 2018-2019 are somewhat updated. How you express in the limitations, if you had included recent seasons the work would have been more interesting.

Response: Dear reviewer#2, thank you so much for these comments and suggestions. We agree with the reviewer respect with the inclusion of recent seasons, and we think that it would be so interesting for the practitioners. However, we have not these data, so we cannot include it.

In my opinion introduction, method section and discussion are correct. You provided a enough content about the context of the manuscript and you have used interesting previous research under the same line that your research. 

I have only suggestion, which I consider that improve your work and will reinforce your results:

Maybe, is it possible to add information about “active” time in the season? This is an important variable. We know that with VAR there are more minutes in the minutes of the discount because a VAR decision could stop the match several minutes. However, in your analysis you don’t include the real “active” time (ball in action) during the matches. I consider that it is interesting if the active time is less with the VAR intervention. Maybe this is one of the reasons which explain that a total distance time and total distance/minutes has been reduced with the VAR apparition. You explain this in the discussion section. Maybe it would be interesting (with your big sample) analyze how affect the VAR to the effective time.

Best regards.

Response: Once again, thank you so much for these comments and your time and constructive comments. We agree with the revisor with the inclusion of the real “active” time (ball in action) during the matches, however we have not these data, so we cannot include it.

Nevertheless, we have included one sentence respect this issue in the "Limitations, and Future Perspectives" section for their inclusion on further studies.

“Finally, due to the VAR there are more time in the minutes of the discount because a VAR decision could stop the match for several minutes (i.e., the real “active” time -ball in action- during the matches decreases), so it would be interesting to analyze how affect the decreased of effective time caused by VAR on match physical demands.”

Reviewer 3 Report

The aim of this paper was to compare two seasons of the La Liga soccer league, one with VAR and one without, on metrics of player match demands assessed via a multi-camera tracking system.

GENERAL COMMENTS

Overall, the paper is clear and concise, with a few modification required prior to publication. There is no major novelty in the paper, with the biggest difference from previous literature being the league selected in the current study, and presentation of high intensity sprint efforts.

Introduction

The introduction presents clear evidence of prior and continuing research on the topic at hand, however, I would suggest the authors provide an additional paragraph explaining why VAR would potentially affect the proposed dependent variables, especially since previous research has not probed into this. The rationale for the current dependent variables (I.e., high intensity distance) is not clear.

Results

The tables and figures are both representing the same data. Please remove either the table or the figures. I would suggest removing the figures., as the table presents all the data in a clear and concise manner.

SPECIFIC COMMENTS

Please find specific comments in the attached PDF file.

Author Response

Reviewer comments

Reviewer 3

 The aim of this paper was to compare two seasons of the La Liga soccer league, one with VAR and one without, on metrics of player match demands assessed via a multi-camera tracking system.

General comments

Overall, the paper is clear and concise, with a few modification required prior to publication. There is no major novelty in the paper, with the biggest difference from previous literature being the league selected in the current study, and presentation of high intensity sprint efforts.

Response: Dear reviewer#3, thank you so much for these comments and your time and constructive criticisms. We think that the comments are suitable for the improvement the quality of our study in the new version.

INTRODUCTION

The introduction presents clear evidence of prior and continuing research on the topic at hand, however, I would suggest the authors provide an additional paragraph explaining why VAR would potentially affect the proposed dependent variables, especially since previous research has not probed into this. The rationale for the current dependent variables (I.e., high intensity distance) is not clear.

Response: Thank you so much for this appraisal. As mentioned before, we have changed the structure and wording of the entire Introduction. In this regard, we have specifically re-written this paragraph as follows:

RESULTS

The tables and figures are both representing the same data. Please remove either the table or the figures. I would suggest removing the figures., as the table presents all the data in a clear and concise manner.

Response: Thank you for your suggestion. Initially, we decided include it to visually facilitate the reader's understanding. However, we agree with the revisor that the table and figures show the same information. In addition, another reviewer has pointed out a similar observation. Therefore, we have removed the figures for the clearer data presentation.

Specific comments

Please find specific comments in the attached PDF file.

Response: Thank you so much for these comments. We agree with the reviewer#3, and they have been addressing throughout the manuscript. Following, we have included the changes addressed in the manuscript:

Comment 1 (L29): We have modified the Introduction section to be clearer and more concise in the study rationale (In page 1).

Comment 2 (L105): Thank you for the observation. We have corrected the mistake and modified the sentence as follows (In page 3):

“This system records and analyzes X, Y and Z positions for each player from several angles, thus providing real-time three-dimensional tracking (tracking data are recorded at 25 Hz).”

Comment 3 (L110): Thank for the observation. We have corrected the mistake (In page 3).

Comment 4 (L121-122): Thank you. We have followed the suggestion. We have modified the sentence as follows (In page 3):

“All these variables show the total sum of the physical performance of the players who participated in each match (i.e., all players who completed entire matches, all players who were replaced, and substitute players).”

Comment 5 (128-131): Thank you so much for the appraisal. We have modified the sentence to be clearer. In addition, we have followed the framework according to previous studies (please see Pons et al., 2021; In page 3).

Pons, E., Ponce-Bordón, J. C., Díaz-García, J., López del Campo, R., Resta, R., Peirau, X., & García-Calvo, T. (2021). A longitudinal exploration of match running performance during a football match in the spanish la liga: A four-season study. International Journal of Environmental Research and Public Health18(3), 1-10.

Round 2

Reviewer 1 Report

Dear authors,

thanks for the revised version of your manuscript. I can clearly see the improvement of it.

However, I've a few lingering comments.

General:

I insist that you let your manuscript be edited by professional native English speakers. There are enterprises that do that for you without being too expensive.
As you have adressed that point by letting it revised by a person that seem to speak propper English, the first sentence of the Intro is grammatically wrong. There is no 'showed' in the perfect tense you need to use the past participle 'shown'. As these are important errors in English language, I please you to let this done professionally. Similar errors occur throughout the mansucript.

Methods:

I've never read to give affilliations as a reference. But if you like I'm fine.

I'm still hesitant if using the term difference, even in the phrasing, "non-significant differences" is correct. From a scientific view: If there is no significance than there is no difference. Other: use the effect size to explain if differing values have the potential getting significant if you would have had higher sample size.

Results:

I still do not see the point including those figures in the manuscript. They do help nothing to improve the readability and understaning of your results. Instead they are rather confusing as still not easy to understand. Consider removing them. Sorry for the work you put in those.

Author Response

Dear Reviewer 1,

We are sending you the letter of response corresponding to the second revision with the pertinent modifications.

King Regards.

LETTER OF RESPONSE

Editor comments

(I) Please revise your manuscript according to the referees’ comments and upload the revised file by 17 April 2022.
(II) Please use the version of your manuscript found at the above link for your revisions.
(III) Any revisions made to the manuscript should be marked up using the “Track Changes” function if you are using MS Word/LaTeX, such that changes can be easily viewed by the editors and reviewers.
(IV) Please provide a short cover letter detailing your changes for the editors’ and referees’ approval.

If one of the referees has suggested that your manuscript should undergo extensive English revisions, please address this issue during revision. We propose that you use one of the editing services listed at
https://www.mdpi.com/authors/english or have your manuscript checked by a native English-speaking colleague.

Please do not hesitate to contact us if you have any questions regarding the revision of your manuscript or if you need more time. We look forward to hearing from you soon.

Response: Firstly, we would like to thank the reviewer#1 and the Editor for their valuable comments and suggestions for our work. All these recommendations have been considered in the current version of the manuscript to improve the quality of the article.

In addition, the manuscript has been revised again by an English-native speaker, although maybe that person was no specialist in this topic. Nevertheless, we have tried to improve the English sentences structure of the manuscript.

Reviewer comments

Reviewer 1

Dear authors,

thanks for the revised version of your manuscript. I can clearly see the improvement of it.

However, I've a few lingering comments.

General:

I insist that you let your manuscript be edited by professional native English speakers. There are enterprises that do that for you without being too expensive. As you have addressed that point by letting it revised by a person that seem to speak proper English, the first sentence of the Intro is grammatically wrong. There is no 'showed' in the perfect tense you need to use the past participle 'shown'. As these are important errors in English language, I please you to let this done professionally. Similar errors occur throughout the manuscript.

General response: Thank you so much to the Reviewer#1 for this comment. The manuscript has been revised again by an English-native speaker, although maybe that person was no specialist in this topic. Nevertheless, we have tried to improve the English sentences structure of the manuscript.

METHODS

I've never read to give affiliations as a reference. But if you like I'm fine.

I'm still hesitant if using the term difference, even in the phrasing, "non-significant differences" is correct. From a scientific view: If there is no significance than there is no difference. Other: use the effect size to explain if differing values have the potential getting significant if you would have had higher sample size.

Response: Thank you so much for the appraisal. We have removed the term “non-significant differences” to be clearer.

Secondly, respect to the effect size, the differences on values have not the potential getting significant if we would have had higher sample size. We included the next information to justify this rationale:

According to Hopkins et al. (2009), we used the threshold values for Cohen’s ES statistics of trivial (< 0.2), small (0.2 – 0.6), moderate (0.6 – 1.2), large (1.2 – 2.0), and very large (> 2.0). In this vein, our values vary between 0.04 - 0.38, so we consider low values. Therefore, our values have not the potential getting significant.

RESULTS

I still do not see the point including those figures in the manuscript. They do help nothing to improve the readability and understanding of your results. Instead, they are rather confusing as still not easy to understand. Consider removing them. Sorry for the work you put in those.

Response: Thank you for the suggestion. We have finally removed the figures on the manuscript.
